# Realization of sextuple polarization states and interstate switching in antiferroelectric CuInP$_2$S$_6$

Tao Li [1,6], Yongyi Wu [1,6], Guoliang Yu [2], Shengxian Li[2], Yifeng Ren[3], Yadong Liu[1], Jiarui Liu[1], Hao Feng[1], Yu Deng[3], Mingxing Chen [2,4]✉, Zhenyu Zhang [5]✉ & Tai Min [1]✉

Realization of higher-order multistates with mutual interstate switching in ferroelectric materials is a perpetual drive for high-density storage devices and beyond-Moore technologies. Here we demonstrate experimentally that anti-ferroelectric van der Waals CuInP$_2$S$_6$ films can be controllably stabilized into double, quadruple, and sextuple polarization states, and a system harboring polarization order of six is also reversibly tunable into order of four or two. Furthermore, for a given polarization order, mutual interstate switching can be achieved via moderate electric field modulation. First-principles studies of CuInP$_2$S$_6$ multilayers help to reveal that the double, quadruple, and sextuple states are attributable to the existence of respective single, double, and triple ferroelectric domains with antiferroelectric interdomain coupling and Cu ion migration. These findings offer appealing platforms for developing multistate ferroelectric devices, while the underlining mechanism is transformative to other non-volatile material systems.

The contemporary explosion of information data drives persistent demands for higher-order multistate non-volatile memory, with the primary objectives of significantly increased storage capacity, improved energy efficiency, and reduced costs[1,2]. In this context, multistate ferroelectric devices, including ferroelectric random-access memory cells, ferroelectric field effect transistors, ferroelectric tunnel junctions, and ferroelectric neural network elements[3–7], have emerged as flourishing research areas in the field of beyond-Moore technologies[8–11]. Earlier efforts in this endeavor were focused on three-dimensional ferroelectric oxides, which nevertheless suffer from the intrinsic thickness limit when exploited in thin film form, making it difficult to achieve higher-order multiple polarization states[12]. In fact,

the number of discrete polarization states experimentally observed so far based on ferroelectric oxides has not exceeded four[13–15]. The recently discovered two-dimensional (2D) ferroelectrics, in principle, can overcome the fundamental scaling problem imposed by the thickness limit, owing to their inherent features of dangling bonds-free surfaces and weak van der Waals (vdW) interlayer couplings[9,12]. Given their distinct merits and advantages, it is natural to expect that new opportunities will emerge to realize higher-order multiple polarization states based on vdW ferroelectric materials, which in turn may find significant non-Boolean nanoelectronic applications.

Indeed, even though the field of vdW ferroelectrics has emerged for only a few years[16–19], significant advances have been made in multi-

[1]Centre for Spintronics and Quantum Systems, State Key Laboratory for Mechanical Behavior of Materials, School of Materials Science and Engineering, Xi'an Jiaotong University, 710049 Xi'an, China. [2]Key Laboratory for Matter Microstructure and Function of Hunan Province, Key Laboratory of Low-Dimensional Quantum Structures and Quantum Control of Ministry of Education, Synergetic Innovation Centre for Quantum Effects and Applications (SICQEA), School of Physics and Electronics, Hunan Normal University, 410081 Changsha, China. [3]Solid State Microstructure National Key Lab and Collaborative Innovation Centre of Advanced Microstructures, Nanjing University, 210093 Nanjing, China. [4]State Key Laboratory of Powder Metallurgy, Central South University, 410083 Changsha, China. [5]International Center for Quantum Design of Functional Materials (ICQD) and Hefei National Laboratory, University of Science and Technology of China, 230026 Hefei, Anhui, China. [6]These authors contributed equally: Tao Li, Yongyi Wu. ✉e-mail: mxchen@hunnu.edu.cn; zhangzy@ustc.edu.cn; tai.min@xjtu.edu.cn

fronts, including achieving higher-order polarization states. The first compelling example is the demonstration of quadruple polarization states in $CuInP_2S_6$–$In_{4/3}P_2S_6$ mixed-phase samples, with the four states corresponding to the multiplication of two Cu ionic locations in the traditional intralayer and two more exotic interlayer configurations[20–22]. The latter two were achieved by invoking Cu ionic displacements, as indicated by first-principles calculations[20–22]. More recently, four polarization states have also been demonstrated using 3R-$MoS_2$ thin films[23] within the context of sliding ferroelectrics[24–27]. These recent exciting and inspiring achievements strongly indicate that it is ripe to aim for even higher objectives of the field, namely, to achieve higher-order (more than four) polarization states based on 2D ferroelectrics, which, if successful, would be of distinct fundamental and apparent technological significance.

Here, we successfully break the status quo on the polarization states of four achieved in either 3D or 2D ferroelectric systems by unambiguously demonstrating robust sextuple polarization states in vdW $CuInP_2S_6$ films. For a given system that harbors a polarization order of six states, we further show that it can also be reversibly tuned into the order of four or two states, and for a given polarization order, mutual interstate switching can be achieved, both via moderate electric field modulation. Guided by first-principles calculations of multi-layered $CuInP_2S_6$ films as model systems, we propose an innovative concerted mechanism to interpret the stabilization of the sextuple polarization states as well as their mutual interstate switching, invoking vertically stacked and antiferroelectrically coupled domains and cross-domain migration of the Cu ions. Collectively, these findings offer appealing platforms for developing multistate ferroelectric devices, while the underlining mechanism is transformative to other non-volatile materials and devices.

## Results and discussion
### Pristine CIPS ferroelectric domains
Our chemical vapor transport (CVT)-grown $CuInP_2S_6$ (CIPS) crystals have demonstrated trigonal crystal structure in P31c space group (Fig. 1a, c) and Cu/In ratio of 0.97 ± 0.02 (Table S1), similar to the CIPS structure reported by Deng et al. [28], but different from the crystals showing quadruple polarization states with a monoclinic structure in Cc space group and a Cu-deficient composition of $Cu_{0.4}In_{1.2}P_2S_6$ (Cu/In ratio of 0.33)[20]. Moreover, X-ray diffraction (XRD) characterizations of our CIPS crystal show a single-crystalline nature with a c-lattice of 13.04 Å (Fig. S1), agreeing well with the trigonal crystal structure (Fig. S2). We further conducted elemental mappings of the cross-section (Fig. S3) and the surface (Fig. S4) of CIPS films by high-resolution transmission electron microscopy (HR-TEM), which revealed the uniform distribution of the four elements without any phase separation. A non-piezoelectric separate phase, named $In_{4/3}P_2S_6$ (IPS), has been reported in many CIPS films[20,29–31], including the CIPS system showing quadruple polarization states. In this CIPS system, the IPS-induced local strain is required to stabilize the interlayer (±$Cu_I$ in Fig. 1b, displaced into the vdW gap) and intralayer (±$Cu_{II}$ in Fig. 1b) Cu positions that produce the four polarization states at room temperature, whose energy difference is only ~14 meV[32]. To further verify the absence of IPS in our crystals, we performed Raman spectroscopy measurements and compared the Raman spectrum between pure CIPS and CIPS-IPS mixed phases. Three strong characteristic peaks (127, 140, and 255 cm$^{-1}$) are clearly identified to be associated with the IPS phase only[33]. In our Raman spectrums (Fig. 1d), two of the IPS characteristic peaks (127 and 255 cm$^{-1}$) are absent, while the 140 cm$^{-1}$ peak has negligible intensity. Therefore, we believe that our crystals are pure CIPS without the IPS phase, thus, the formation and stabilization of the four polarization

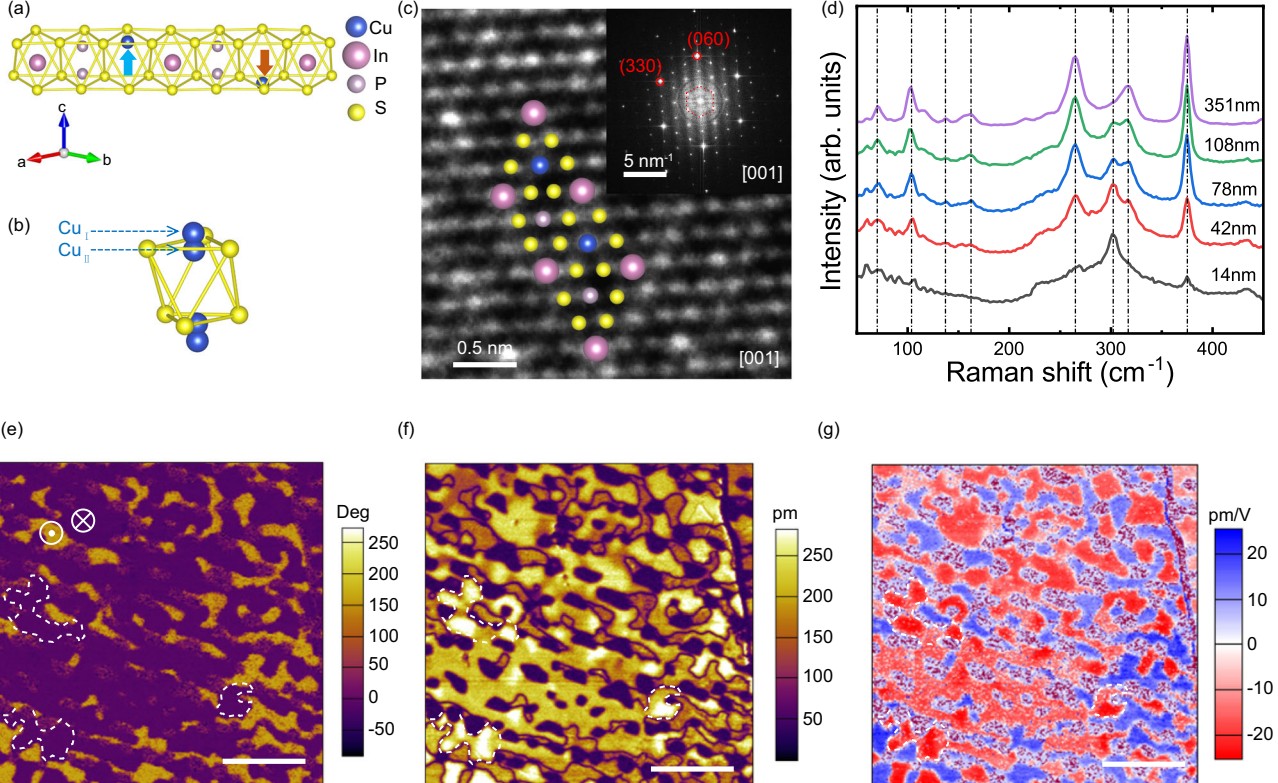

**Fig. 1 | Crystal structure and pristine domain pattern of CIPS films. a** Atomic structure of monolayer (ML) CIPS with the trigonal crystal structure. Cu, In, P, and S atoms are labeled accordingly. **b** Possible Cu ion occupation sites in a sulfur octahedral frame below Curie temperature. **c** HR-TEM image and fast Fourier transform (FFT) pattern of CIPS film in the direction perpendicular to the basal plane of CIPS. Cu, In, P, and S atoms are placed at corresponding positions for the visual guide. **d** Raman spectroscopy of mechanically exfoliated CIPS films with different thicknesses. **e** PFM phase, **f** amplitude, and **g** piezoresponse images reveal the pristine domain pattern of a CIPS film. The scale bar is 1 μm.

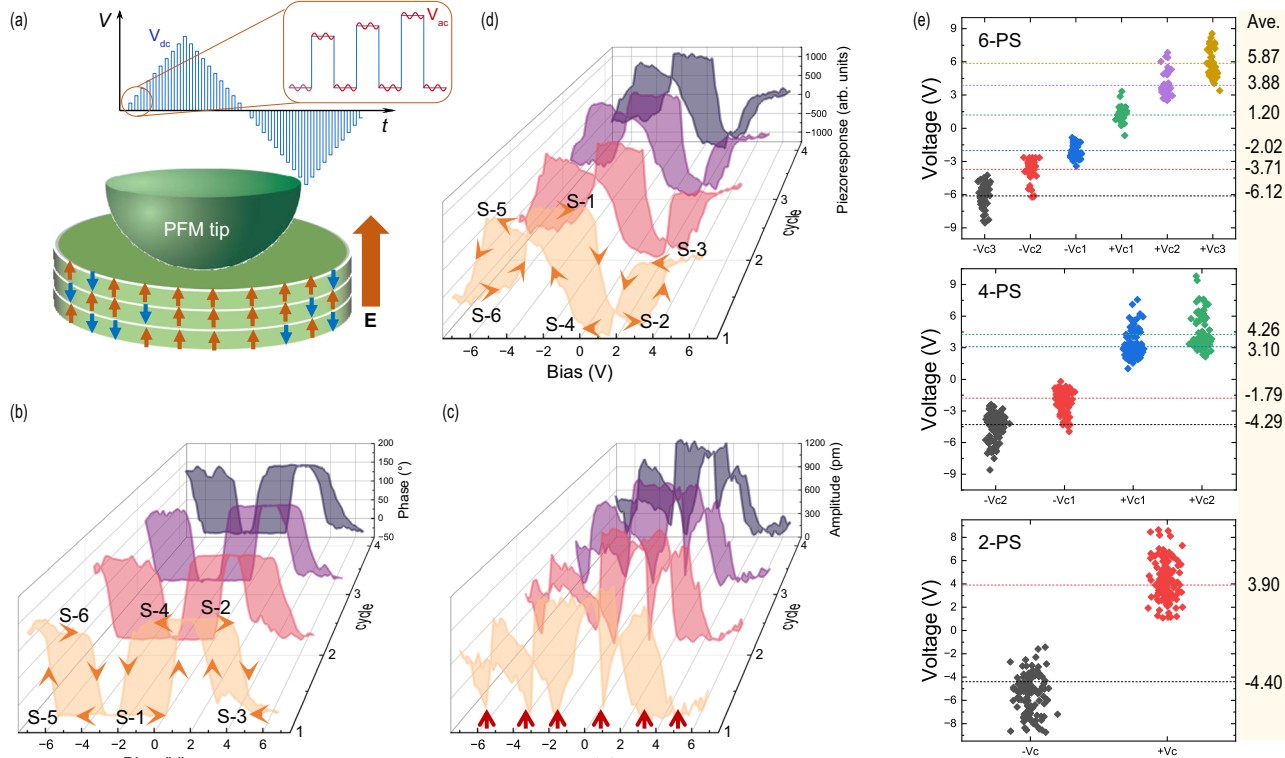

**Fig. 2 | Sextuple polarization states and corresponding coercive voltage distribution. a** Schematic diagram of the PFM measurement setup. Voltage is applied through the conductive PFM tip, and the substrate is grounded. The electric field direction is indicated by a fat arrow on the right of the sample. The up and down arrows in the film area represent the upward and downward polarization dipoles, respectively, which help to illustrate the non-uniform dipole distribution under the tip. The inset shows the applied DC pulsed triangular square wave ($V_{dc}$) embedded with an AC detection signal ($V_{ac}$) in an SS-PFM sweeping cycle. **b** Phase, **c** amplitude, and **d** piezoresponse hysteresis loops of the CIPS sample showing sextuple polarization states. Four cycles of hysteresis loops measured at the same location are plotted along the cycle axis in different colors. Six polarization states are labeled on **b** and **d** as S-1–S-6, and arrows on the loops indicate the switching path. Coercive voltages ($V_c$) are identified by the valleys in a representative amplitude loop, as indicated by the red arrows in (**c**). **e** Distribution of $V_c$ for different polarization orders (6-, 4- and 2-PS, polarization states), with the average value of each state labeled on the right.

states (±$Cu_I$ and ±$Cu_{II}$ positions) at room temperature via the IPS phase-induced strain is not applicable in our case.

To evaluate the pristine ferroelectric domain structure in our CIPS films in the out-of-plane direction, we used the resonance-enhanced piezoresponse force microscopy (PFM) technique[34]. All of our experiments and analyses are focused on sample areas that showed strong amplitude responses (85%), away from the regions with weak amplitude that are close to the noise level of the PFM system (dark spots in Fig. 1f and Fig. S5). The pristine CIPS film exhibits a multidomain pattern as shown in the phase image (Fig. 1e) and varied contrast in the amplitude image (Fig. 1f). In PFM measurements, a uniform phase contrast indicates the same polarization orientation, these domains usually have a uniform amplitude level in the sensible range of PFM indicating similar polarization strength. However, although some areas show the same phase contrast (purple downward domains enclosed by the dashed white lines in Fig. 1e), the corresponding amplitude can demonstrate multiple levels (Fig. 1f). Figure 1g also clearly demonstrates the variation of piezoresponse strength with the same polarization orientation of these areas. Such observation suggests that the domain with a lower amplitude level is not completely in the downward direction in depth, meaning that domains with the opposite polarization (upward) or even antiferroelectric (AFE) configuration may present in the subsurface.

## Switchable sextuple polarization states

We investigated the domain switching dynamics of the CIPS films using the switching spectroscopy PFM (SS-PFM) technique[35] (see the "Methods" section). In the measured SS-PFM loop, the relative ~180°

phase difference indicates the polarization switching between upward and downward directions, and the amplitude reflects the strength of average piezoresponse within the detection range of PFM. Upon the SS-PFM sweeping (Fig. 2a), we have consistently observed the remanent piezoresponse hysteresis loops (off-field, $V_{dc} = 0$) comprising unprecedented triple-subloop in CIPS films (Fig. 2d). The corresponding phase signal has been reversed six times by ~180° between subsequent switched states from state-1 (S-1) to state-6 (S-6) (Fig. 2b) and the amplitude signal shows corresponding six minima at each polarization switching voltage, i.e., the coercive voltage ($V_c$) (marked by red arrows in Fig. 2c). This phenomenon has been observed consistently over 23 CIPS samples from various crystal batches (an additional dataset is present in Fig. S16), which indicates the robustness of the observed sextuple polarization states. Moreover, we also obtained piezoresponse hysteresis loops with double-subloop (Fig. S6) and regular single-loop (Fig. S7) in our CIPS samples, which correspond to the quadruple and double polarization states, agreeing with the observed number of states reported in many previous works[16,20,32,36–41]. Here we note that the multiple polarization states as established here are fundamentally different from the multiple resistive states reported in previous studies of ferroelectric synapses[6,8,42–44], as distinctly contrasted by the observations of mutually convertible six/four/two polarization states in a single hysteresis loop in the former case, and only two polarization states in the latter case. For the different polarization orders with sextuple, quadruple, and double polarization states, the distributions of respective $V_c$ were plotted in Fig. 2e. We note that there are large variations in the coercive voltages for different polarization states, which can be attributed to the different types

of local defects and active Cu movements at room temperature. In addition, the data presented in Fig. 2e were compiled from our comprehensive dataset without discriminating between crystal batches, film thicknesses, measurement locations, or ranges of bias windows. It is noteworthy that the variation of $V_c$ is much narrower based on the hysteresis loops measured on one sample (Table S3) compared to Fig. 2e. Moreover, ferroelectric domains of CIPS are known to be highly sensitive to strain modulation[45], for example, the aforementioned quadruple polarization states could be stabilized by the local strain from IPS phase in CIPS[20]. To analyze the strain effect in our hysteresis measurements, we applied varied loading forces during the measurements and found that the number of switchable states is not sensitive to the loading force up to about 480 nN (Fig. S8). Additionally, the stability of the sextuple remanent states has been investigated with varied writing/reading pulse duration (Fig. S17), suggesting that the variations in the time scale of 15–50 ms do not alter the major characteristics of the observed sextuple polarization states. Moreover, appropriate detection voltage is necessary to observe reliable sextuple polarization states. The excessive detection voltage can lead to poor repeatability and disrupt the intermediate states, which may cause the states to collapse into more stable states with deeper potential wells (Fig. S19).

A consistent striking experimental observation in the CIPS films exhibiting sextuple or quadruple polarization states is that under relatively strong electric fields (E-fields), the remanent polarization direction is always opposite to the E-field ($P \uparrow \downarrow E$). The phenomenon of polarization opposing the E-field was initially observed by Neumayer et al. in Ni/CIPS/Ni capacitors[21], yet under specific electric pulse conditions, $P \uparrow \downarrow E$ only occurred in partial regions, while the polarization in the other areas still aligned with the E-field ($P \uparrow \uparrow E$). One plausible mechanism to explain the behavior of $P \uparrow \downarrow E$ is linked to the interlayer migration of Cu ions crossing the vdW gap, as originally proposed by Neum1ayer et al.[21] and O'Hara et al.[22], but with a distinct modification: here, the Cu ions can migrate across domains. Notably, in CIPS films showing higher-order polarization states (sextuple), the $P \uparrow \downarrow E$ was always observed in the two states excited by the strongest E-fields (S-3 and S-6 in Fig. 2d), consistently appeared in our hysteresis measurements conducted on various samples. The identical case of $P \uparrow \downarrow E$ observed in the sextuple polarization states also appeared in our CIPS films that demonstrate quadruple polarization states.

## Controllable mutual transformation among polarization orders

In addition to the previous observations of polarization orders with sextuple, quadruple, and double polarization states, we also made a striking observation that these polarization orders can be readily transformed back and forth ($6 \rightarrow 4 \rightarrow 2 \rightarrow 4 \rightarrow 6$) by tuning the bias window for hysteresis loop measurements in the same CIPS film (Fig. 3). Additionally, we also can achieve the polarization order conversions between quadruple and double polarization states ($4 \rightarrow 2 \rightarrow 4$) in CIPS films that originally show quadruple polarization states (Fig. S9). These observations are significantly different from the traditional experimentally reported ferroelectric materials that typically possess a fixed number of intrinsic polarization states, either two[9,46], three[13,14], or four[20], while the number of polarization states or polarization order cannot be changed in a given material system. The unique reversible conversion compellingly demonstrates the tunability of the polarization orders in our CIPS crystals and provides an additional degree of freedom for modulating polarization states. We note that the hysteresis loops observed before and after the reversible transformation are not exactly the same, which can be due to various unavoidable defects in the CVT-grown CIPS films during the deposition and subsequent transfer processes, such as local variations in stoichiometry, vacancies, stacking faults, etc. For example, variations in stoichiometry can lead to local phase separation, which induces local strain and helps to stabilize the quadruple polarization states as proposed in ref. 20. Cu

vacancies and excess Cu ions have been proposed to have the effect of lowering the energy barrier of Cu to move across the vdW gaps[21]. Stacking faults can lead to local kink, edge-type, or knot-type dislocations, which can cause the formation of nanodomains in CIPS films[47]. Nevertheless, even though the polarization switching details, such as the coercive voltage and polarization value, can be varied because of the presence of the aforementioned defects, the polarization orders can be distinguished and transformed within an acceptable tolerance. It is also known that defects can pin the domain boundaries and augment the energy barrier associated with polarization switching. For CIPS systems, the characteristic time associated with the domain wall pinning and depinning from defects have not been experimentally observed yet. According to a quantum-molecular-dynamics result that suggests Cu movement time of ~100 ps across a vdW gap in CIPS with excess Cu[21], we conjecture that our applied electric pulse time (in the millisecond range as described in the Method section) considerably exceeds the characteristic times of the pinning and de-pinning from defects in CIPS.

Among the hysteresis loop measurements over about one hundred samples, the thinnest CIPS films exhibiting double, quadruple, and sextuple polarization states have respective thicknesses of 9.1, 13.6, and 24.2 nm, corresponding to ~12, ~20, and ~35 MLs based on ~6.72 Å thickness of a monolayer (ML) CIPS with a ~3 Å vdW gap[32]. In the 9.1 nm CIPS film, only two polarization states were observed, indicating a uniform polarization switching driven at this film thickness. On the other hand, the quadruple and sextuple polarization states were only observed in thicker films (13.6 and 24.2 nm), suggesting that complex remanent domain configurations in the subsurface can be induced, possibly by the nonuniform E-field excitation under the tip. The quadruple polarization states can be well explained by an existing mechanism proposed recently[20], however, an additional metastable Cu position is needed to explain our sextuple polarization states, suggesting a distinct mechanism from that proposed in ref. 20. To interpret these observations, we appeal to a recent first-principles model study, which proposed an alternative mechanism that forms the quadruple and sextuple polarization states in pure CIPS[48]. Briefly, the quadruple/sextuple states can be obtained via the vertical staking of two/three MLs of CIPS, and each ML can have either intralayer ferroelectric (FE) or antiferroelectric (AFE) domain with AFE/FE interlayer couplings (named AFE/FE model in short). Such a mechanism, once properly generalized by including the Cu ion migration, can qualitatively and semi-quantitatively explain our present observations, as presented later surrounding Fig. 4.

## Mechanism of sextuple-polarization-state

The sextuple polarization states can be conceived by the vertical stacking of three layers of AFE/FE domains with interdomain couplings, as in the 3-ML CIPS system suggested by the AFE/FE model. However, our CIPS films exhibiting sextuple polarization states are much thicker than 3-ML, which necessitates the generalization from AFE/FE MLs to AFE/FE domain blocks (DBs). Here, each DB behaves more-or-less in unison and interacts with neighboring DBs through inter-DB AFE/FE coupling, analogous to the monolayered AFE/FE domain studied in the first-principles calculations. We posit that the defects with no net polarization contribution in CIPS, such as excess Cu ions, Cu vacancies, and random AFE domain spots, can initiate the boundaries between the DBs, which work as dead interfacial layers. The DB is assumed as an energetically metastable elementary unit that effectively acts like an ML in the AFE/FE model and can be collectively modulated by a moderate E-field (as exemplified in Fig. 4c). Similarly, the formation of quadruple polarization states requires two DBs, akin to 2-ML domains in the AFE/FE model system, while double polarization states can be achieved with just one DB. The construction rules of a DB based on the thinnest CIPS films demonstrated double,

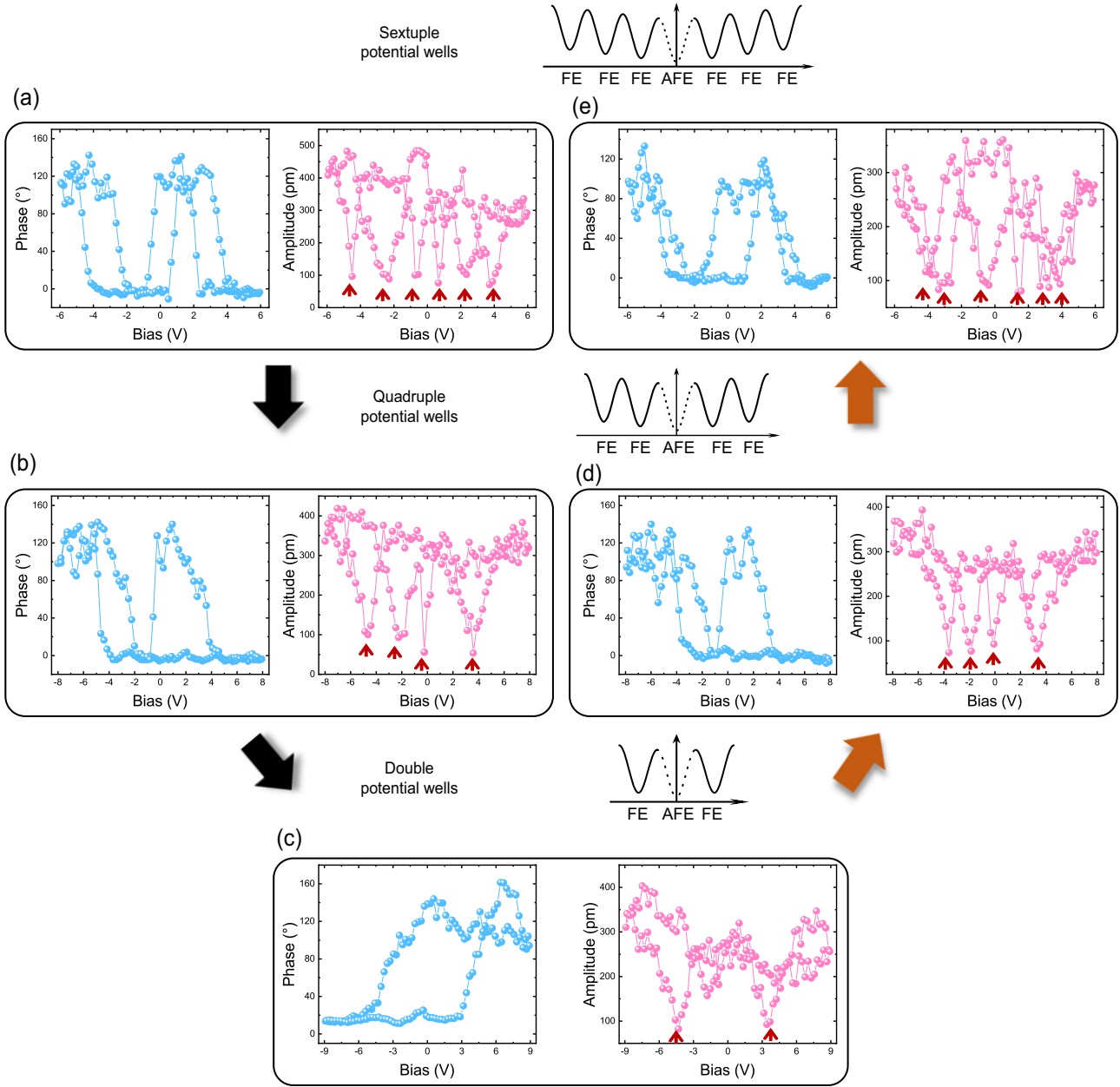

**Fig. 3 | Reversible tuning of polarization orders via E-field modulation on a CIPS film. a–e** Phase (Pha) and amplitude (Amp) hysteresis loops showing different polarization orders. The bold arrows indicated the reversible transformation path. **a** The CIPS film initially exhibited a polarization order of sextuple states obtained within a relatively low bias window of $\pm 6V_{dc}$. **b** When the bias window is increased to $\pm 8V_{dc}$, the order of sextuple states is converted to four. **c** With the bias window further increased to $\pm 9V_{dc}$, it is further converted to the order of two. The order of double polarization states is recovered to the orders of quadruple **d** and the sextuple **e** polarization states by decreasing the bias window to $\pm 8V_{dc}$ and $\pm 6V_{dc}$ subsequently.

quadruple, and sextuple polarization states within the framework of the AFE/FE model are described in Supplementary Note 1.

Based on the ingredient of AFE/FE DBs, alongside the existing Cu ion migration mechanism required to elucidate the observed $\mathbf{P} \uparrow \downarrow \mathbf{E}$ phenomenon, we propose a plausible hybrid mechanism to explain the observed reversible switching of sextuple polarization states from S-1 to S-6 (Figs. 2d and 4b), as well as possible pathways for interstate switching. Our DFT calculations of the 3-ML CIPS system provide insight into the energy barriers among sextuple polarization states, falling within the range of 283–393 meV[48]. Based on the well-known compensation effect in activation processes, we speculate that the energy barriers for DB switching are in the range larger than the maximum energy barrier (393 meV) but smaller than the sum of them (676 meV) with lowered attempt frequency. The proposed mechanism

is further corroborated by the temperature-dependent experiments (Fig. S20). At lower temperatures (5 °C), the energy barrier heights for the sextuple polarization states are too high, and the external electric field can only facilitate Cu ions to overcome the two most shallow wells, thereby resulting in double polarization states only. At elevated temperatures (between room temperature and Curie temperature), the energy barrier heights for sextuple polarization states are readily overcome, which may be due to more active Cu ions movement, resulting in the emergence of sextuple polarization states.

Averaged over 23 experimental datasets, we identified four remanent piezoresponse levels (exemplified in Fig. 4b), which are defined as LH-1(level high, for S-1 and S-4) in the small E-field range, LH-2 (S-2 and S-5) in the medium E-field range, and LL-1 (level low, for S-3) and LL-2 (S-6) in the large E-field range. A critical point to be noted is

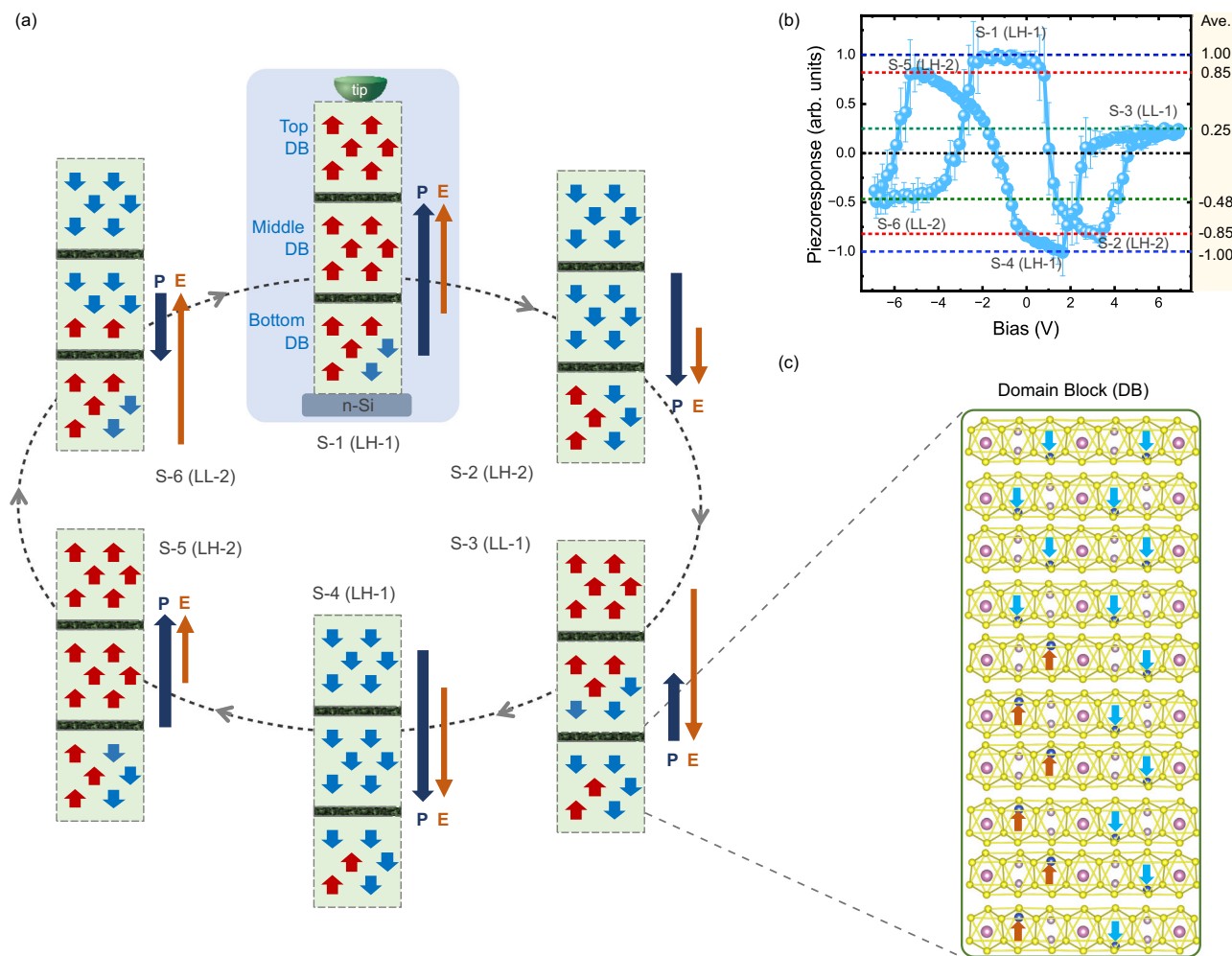

**Fig. 4 | Illustration of a plausible scheme for domain evolution of switchable sextuple polarization states based on the hybrid mechanism. a** Formation of sextuple polarization states from S-1 to S-6 upon DC poling pulses sweeping. The corresponding net piezoresponse (**P**, blue) and electric field (**E**, red) showing relative direction and strength are labeled for each state. Three DBs (top, middle, and bottom) are labeled in S-1. The top DB is right under the tip and the bottom DB is close to the substrate. **b** Representative normalized piezoresponse hysteresis loop obtained from a 90 nm CIPS film, labeled with six polarization states and corresponding four piezoresponse levels. The error bars are calculated based on four cycles of hysteresis loops measured at the same location. **c** Exemplified 10-ML DB with corresponding complex AFE/FE ML configurations.

that the tip-induced E-field is highly nonuniform and decays exponentially with depth in the CIPS film (Fig. S11). This nonuniformity exerts different effects on the vertically stacked DBs (top, middle, and bottom) that form the sextuple polarization states. The impact of different tip sizes has also been discussed in Supplementary Note 3. The top DB experiences the most intense E-field among the three, and under a sufficiently large field, migration of Cu ions across the vdW gaps can be triggered, resulting in a polarization direction opposing the applied E-field[21,22]. For middle DB, depending on the exact strength of the E-field, the Cu ions migration can cross domains and occur in part of the middle DB. At moderate field strength, regular polarization switching can occur in all three DBs from an upward to a downward direction and vice versa. However, in certain states (e.g., S-1 → S-2), the E-field is insufficient to induce the regular polarization switching, especially in the bottom DB that experiences the weakest E-field. Instead, an intermediate AFE state can be induced, aligning with the energy-favorable half-layer-by-half-layer switching mechanism as proposed in the AFE/FE model (Fig. S13 and path-B in Fig. S14). Incorporating the contributions of the three DBs, the schematic showing relative changes in polarization direction and strength during the reversible switching cycle from S-1 to S-6 is depicted in Fig. 4a, the details of which are delineated in Supplementary Note 2 and Fig. S15.

The polarization levels proposed by the hybrid mechanism (Fig. 4a) well match the experimentally observed four piezoresponse levels (Fig. 4b) with similar ratios among the four (Table S2), which lends further support to its validity using vertically stacked DBs as an explanation for the sextuple polarization states.

Furthermore, the mutual transformation among different polarization orders is predominantly influenced by the bias window, representing the maximum strength of the electric field serving as an initializing field for the polarization orders. As the maximum field strength increases from that required for the sextuple to the quadruple/double states, we anticipate that a more uniform domain structure can be produced and more defects can be driven to move around. Upon sweeping the electric bias repeatedly, the tip-induced nonuniform electric field can produce the corresponding domain structures that form the quadruple/double polarization states. Conversely, electric field sweeping with a reduction in the maximum field strength disturbs the uniform domain structure and the defects, leading to the emergence of sextuple polarization states again.

In this work, we grew CIPS (Cu/In ≈ 0.97) crystals showing trigonal crystal structure and P31c space group without IPS phase. In the CIPS films exfoliated from these crystals, we realized the switchable sextuple polarization states and also observed an unusual phenomenon of

remanent polarization direction opposite to an electric field direction after a reasonably strong electric field. The sextuple polarization states can be elucidated by the vertically stacked complex AFE/FE domain blocks and their interactions with ionic movement crossing the domain. In addition, the interconversions among different polarization orders were also observed simply by varying the bias window, which enables an extra degree of freedom to modulate the ferroelectric devices. 2D ferroelectric materials like CIPS having intrinsic multistates are highly desirable for multi-level non-volatile memory devices due to the potential simple fabrication process and long endurance. Our findings offer appealing layered materials platforms for developing multistate ferroelectric devices, while the underlining mechanism is transformative to other non-volatile material systems.

## Methods

### Crystal growth

Single crystalline CIPS crystals were synthesized by the chemical vapor transport (CVT) method. The powders of Cu (99.80%), In (99.99%), P (98.90%), and S (99.90%) were encapsulated in a vacuum quartz tube (vacuum at the level of $\sim 10^{-3}$ Pa) according to stoichiometric ratios, and $I_2$ was used as the transport agent at a dosage of 2 mg/cm$^3$. The temperatures of the feedstock zone and the crystal growth zone were kept at 750 °C and 650 °C, respectively. After 7 days of reaction under these conditions, followed by a natural cooling process, greenish-yellow crystals were obtained.

### Crystal structure and composition characterizations

The CIPS crystal structure was characterized using an X-ray diffractometer (D/MAX-2400, Rigaku, Japan). A Cu target was selected as the X-ray emission source and data were collected in the range of 10°–80° with a scanning speed of 5°min$^{-1}$. For compositional characterization, the freshly dissociated crystals were mechanically exfoliated by a conductive carbon adhesive tape, and a clean and flat surface was used for scanning electron microscopy (Sigma 300, Zeiss, Germany) and analyzed for elemental distribution. The corresponding atomic percentages in the region were obtained using energy dispersive spectroscopy (EDS) (Xplore30, Oxford Instruments, USA). In addition, Raman spectra of CIPS films of different thicknesses were collected using a Renishaw Raman microscope (InVia Qontor, Renishaw, UK) with an incident excitation light source at 523 nm.

### High-resolution transmission electron microscopy (HR-TEM)

TEM and EDS experiments were carried out using an FEI Titan 60-300 microscope equipped with a monochromator. The accelerating voltage was 300 kV and the lens aberrations were listed below: two-fold astigmatism A1 < 5 nm, three-fold astigmatism A2 < 20 nm, and axis coma B2 < 10 nm. HR-TEM images were collected using a Gatan One-view camera with a resolution of $4k \times 4k$. To avoid strong ion-beam damage during the early FIB (focused ion beam) preparation period with strong beam currents, the samples were coated with protective Pt layers before TEM and EDS measurements.

### Piezoresponse force microscopy (PFM) imaging

CIPS thin films were transferred on heavily doped n-type silicon substrates by mechanical exfoliation using blue adhesive tape (1007 R, Ultron Systems Inc, USA). Piezoresponse force microscopy (PFM) was conducted on thin flakes of CIPS using a commercial atomic force microscope (MFP-3D, Oxford Instruments, USA) in the atmospheric environment. The Dual AC Resonance Tracking (DART) mode is used to enhance the signal-to-noise ratio and compensate for topographic crosstalk[34]. The Pt/Ir-coated conductive probes (PPP-EFM, Nanosensors, Netherland) with a nominal spring constant of ~3 N m$^{-1}$ and a typical free resonance of 75 kHz were used for signal detection. The AC detection voltages (0.3–0.5 $V_{ac}$) were applied via the conductive probe to the samples at the tip-sample contact resonant frequency

(~350 kHz). The nominal radius of curvature of the tip is 25 nm. The spring constant of the cantilever and the sensitivity of the photo-detector were calibrated using the Sader and thermal noise method[49]. In addition, the raw data of amplitude and phase is fitted using a simple harmonic oscillation model to calculate the piezoresponse[50].

### Switching spectroscopy PFM (SS-PFM)

The localized hysteresis loops of CIPS were collected using the Switching Spectroscopy PFM (SS-PFM) mode by the same PFM tip used for PFM imaging. Pulsed triangular DC driving voltage ($V_{dc}$) was used (Fig. 2a) to modulate the CIPS domains. The DC pulse width is set to 10–15 ms, and the rise time of each pulse is fixed at 0.5 ms. To avoid the alteration of polarization states by the AC detection voltage, we used the detection voltages in the range from 0.3 $V_{ac}$ to 0.5 $V_{ac}$ to detect the piezoresponse signal during and after each DC pulse to ensure the reliability of the measured data. To void the ambiguity from the electrostatic force for on-field (DC voltage is on) measurements, we focused on the off-field (remanent) data for our analyses. Multiple cycles of hysteresis loops were collected for each measurement at a specific location. The coercive fields $V_c$ collected from PFM hysteresis loops were averaged over about 70 measurements for the double, quadruple, and sextuple polarization states, respectively.

### COMSOL multiphysics simulation

The local three-dimensional electric field distribution inside CIPS film is modeled using the static electric equation in the AC/DC module of COMSOL Multiphysics. Two terminals were considered, namely the hemispherical conductive tip and heavily doped Si substrate, where an external voltage $V_{dc}$ was applied to the tip, and the substrate was grounded. The tip radius was set to 25 nm. A Dirichlet boundary condition was imposed, assuming that the air potential infinitely far away from the tip was zero. The relative dielectric constants of CIPS and air were respectively set to 30 and 1.

### Computational method

A machine-learning potential generated from the deep potential method is used to model the interatomic interactions for both the bulk phase and thin films of CIPS[51]. The dataset was generated using the deep potential generator (DP-GEN), which contains information about various configurations of the bulk phase, monolayer, bilayer, trilayer, and quadlayer[52]. DeePMD-kit was used for the training[53]. The involved DFT calculations were performed using the Vienna Ab initio Simulation Package[54]. Polarization states that combine inter- and intra-layer ferroelectric and antiferroelectric couplings were considered during the DFT calculations. The pseudopotentials were constructed by the projector augmented wave method[55]. For the bulk, a $7 \times 4 \times 3$ Γ-centered $k$-mesh was used to sample the Brillouin zone (BZ) for the structural relaxation. For the thin films, a $7 \times 4 \times 1$ $k$-mesh was used to model the 2D BZ. An energy cut-off of 500 eV was used for the plane waves for all the DFT calculations. A 20 Å vacuum region between adjacent plates was used to avoid the artificial interaction between neighboring periodic images for the calculations of the thin films. Van der Waals dispersion forces between the layers were accounted for by the DFT-D3 method[56]. The systems were fully relaxed until the residual force on each atom was less than 0.01 eV/Å. The total energies of all polarization states for 4-ML and 6-ML CIPS were obtained using the machine-learning potential. The energies and atomic forces predicted by DP method for all configurations in the test dataset (Fig. S18) agree well with those of DFT, suggesting the high accuracy of the DP results.

## Data availability

All data generated and analyzed in this study are included in the article and the Supplementary Information. Further datasets are also available from the corresponding author upon request.

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

## Acknowledgements

T.L. acknowledges the support from the National Key R&D Program of China (Grant No. 2021YFA1202200). T.M. acknowledges the support from the National Key R&D Program of China (Grant No. 2022YFB4400200) and the Major Key Project of Peng Cheng Laboratory (Grant No. PCL2023AS1-2). Z.Z. acknowledges the support from the Innovation Program for Quantum Science and Technology (Grant No. 2021ZD0302800), Anhui Initiative in Quantum Information Technologies (Grant No. AHY170000), and National Natural Science Foundation of China (Grant No. 11974323). M.C. acknowledges the support from the National Natural Science Foundation of China (Grant No. 12174098) and the State Key Laboratory of Powder Metallurgy, Central South University, Changsha, China, as well as the High-Performance Computing Platform of Hunan Normal University, where calculations were carried out in part using their computing resources. Y.D. acknowledges the support from the National Natural Science Foundation of China (Grant No. 12274202).

## Author contributions

Y.W., J.L., and H.F. fabricated the CIPS samples, performed the PFM and SS-PFM measurements, and analyzed the data under the guidance of T.L. S.L. and G.Y. performed the first-principles calculations under the guidance of M.C. Y.R. performed the TEM measurements under the guidance of Y.D. Y.D.L. conducted the COMSOL simulations. T.L. drafted the manuscript. Z.Z. and T.M. advised on all efforts. All authors contributed to the discussions and production of the manuscript.

## Competing interests

The authors declare no competing interests.
