## [Peer Review File · Nature Communications]

Realization of sextuple polarization states and interstate switching in antiferroelectric CuInP_2S_6REVIEWER COMMENTS

Reviewer #1 (Remarks to the Author):

This manuscript reports that van der Waals CuInP_2S_6 is able to be stabilized into double, quadruple, and sextuple polarization states. These polarization orders can be switched by tuning the bias window for hysteresis loop measurements. Combining theoretical approaches, the authors attribute the existence of multiple states to complex domain blocks. Evidence for the mechanism is not strong enough. The multiple states observed here are not particularly intriguing, given that more states have already been realized in the field of ferroelectric synapses. Although it might offer a slight improvement in understanding the well-known 2D system, this work is not convincing enough to warrant publication in Nature Communications.

I have some questions or comments for the authors:

1. The authors mentioned that it is possible to tune polarization orders (Fig.3 and Fig.S9). However, the hysteresis loops appear to distinguish between the original (Fig.3a) and recovered (Fig. 3e) sextuple states. Quadruple cases (Fig. 3b .vs. Fig. 3d; Fig. S9a .vs. Fig. S9c) also exhibit this behavior. I thus doubt whether the polarization states belonging to the initial and subsequent polarization orders are exactly the same ones.
2. The off-field amplitude hysteresis loops in Fig. 2c around 4-6 V bias window show that the number of minima varies across cycles. The individual stability and ability to retain after withdrawing electric fields of each sextuple polarization state should be carefully discussed including the time scale.
3. Regarding the mechanism of the sextuple-polarization state, the authors attribute it to different polarization configurations depending on the applied electric field. If so, why are there sextuple states rather than other numbers? The authors indicated in Fig. S12 that some configurations have relative energy levels. To effectively demonstrate this point, all polarization configurations should be taken into account and divided into six groups. In this regard, more theoretical calculations are required.
4. The mechanism of the quadruple polarization states has been well explained in ref.20. Both the HP and LP states share the same polarization arrangements. Their difference is caused by the absence or presence of displacement in Cu sites. To my understanding, the mechanism involving the change in polarization configurations mentioned here is distinct from that. I wonder why did they observe comparable hysteresis loops?
5. The reliability of machine-learning potential should be assessed in advance. It is best to provide a force and energy comparison between DFT and DeepMD. Considering that 4-ML CIPS falls inside the DFT range, I suggest combining DFT and machine-learning potential to compare the total energies of all polarization states for 4-ML CIPS.

Reviewer #2 (Remarks to the Author):

In this work, the authors experimentally demonstrated that antiferroelectric van der Waals CuInP_2S_6 films can be stabilized into various polarization states, including double, quadruple, and sextuple polarization states. They also showed that a system with a polarization order of six can be reversibly tuned to an order of four or two, providing a way to control the number of polarization states.

While the experimental results are interesting and significant, I have several concerns over the explanation and interpretation of the results and the proposed underlying switching

mechanisms. Thus, I would like to seek clarifications from the authors.

1. In CVD-grown CIPS, various defects can occur during the deposition and subsequent transfer processes, for example, local variations in stoichiometry, vacancies, stacking faults, layer thickness variations, surface defects, and contaminants. These defects can significantly impact the ferroelectric domains and their switching behavior. It would be insightful if the authors can assess the impacts of these defects on the domain behavior.

2. It is known that defects can pin the domain boundaries and control the domain boundary movements. I am wondering how the characteristic times of these pinning and de-pinning processes compare with the DC pulse width and the rise time of each pulse. It would be insightful if the authors can discuss these issues.

3. There are large variations in the coercive voltages for different polarization states (Figure 2(e)). The authors should discuss the physical origins.

4. It seems that the authors used ML models to explain quadruple and sextuple polarization states in pure CIPS (the reference is not available for the reviewer). However, the authors used DBs (~ 10 MLs) to explain these multiple polarization states here. This change has not been properly justified. In my opinion, this is a significant weakness of the present work.

5. The discussion on the DBs and their boundary structures and dynamics is absent. It would be insightful if the authors can discuss these issues. Just curious, are the DB boundaries associated with defects?

6. What are the energy barriers between these different states? Can the detection voltages or thermal fluctuations affect the state stability?

7. The authors speculated that the quadruple and sextuple states were possibly due to the nonuniform electric field under the tip. If this is the case, the tip size plays a crucial role in the distribution of the electric field under the tip. But the impact of the tip size was not discussed. If the tip radius decreases from the current 30 nm to, for example, 5 nm, what are the consequences?

Responses to Reviewers' Reports
(MS # NCOMMS-23-52264A by Tao Li et al.)

We thank the two reviewers for their careful and expertized review of the above manuscript. The detailed responses are listed below, and the manuscript has been revised accordingly.

Detailed responses to Reviewer 1

Generic Comments: *This manuscript reports that van der Waals CuInP2S6 is able to be stabilized into double, quadruple, and sextuple polarization states. These polarization orders can be switched by tuning the bias window for hysteresis loop measurements. Combining theoretical approaches, the authors attribute the existence of multiple states to complex domain blocks. Evidence for the mechanism is not strong enough. The multiple states observed here are not particularly intriguing, given that more states have already been realized in the field of ferroelectric synapses. Although it might offer a slight improvement in understanding the well-known 2D system, this work is not convincing enough to warrant publication in Nature Communications.*

Response: We apologize for the likely confusion between multiple resistive states in the field of ferroelectric synapses and the multiple polarization states stressed in the present manuscript, and appreciate this opportunity to better elucidate the distinction between the two. In essence, the multiple states highlighted in the context of ferroelectric synapses are resistive states rather than intrinsic multiple polarization states. The key distinction between the two is that the former case is evidenced by the characteristic double-state ferroelectric hysteresis loops in those ferroelectric synapses [1~5]. Achieving resistive multistate involves adjusting the areal ratio of P_{up} and P_{down} domains in a ferroelectric film or stacking multiple layers of ferroelectric films separated by dielectrics, which can occur in many systems with two intrinsic polarization states, as exemplified in Fig. R1 [1].

Figure R1. Exemplified ferroelectric synapse based on an Ag/PZT/Nd-SrTiO₃ ferroelectric tunnel junction with 256 conductance states. Nevertheless, the ferroelectric hysteresis loops, including phase (a) and amplitude (b), observed by PFM using different detection biases demonstrate double polarization states only, P_{up} and P_{down} [1]. Those loop characteristics are qualitatively similar to our observed double-state hysteresis loop in CIPS by PFM, but fundamentally different from those of the quadruple and sextuple polarization hysteresis loops in the present study.

In contrast, in our CIPS systems, we demonstrated six-fold reversals of polarization directions with six observed corresponding coercive voltages in a single hysteresis loop, which directly attests to

the existence of sextuple polarization states. Besides, we observe double-state hysteresis loops in the CIPS films that originally demonstrated sextuple-state, as well as quadruple-state hysteresis loops from the film that demonstrated sextuple-state initially, representing the same maximum number of intrinsic polarization states experimentally reported so far in the literature, CIPS (Ref 20), 3R-MoS₂ (Ref 23) and non-vdW perovskites (Ref 13~15).

Thus, the *intrinsic ferroelectric multistates* observed in our CIPS samples are fundamentally different from the *resistive multistates* observed in the ferroelectric synapses. Our reported CIPS, featuring sextuple polarization states, also stands as the maximum number of intrinsic polarization states reported so far. More interestingly, the number of switchable polarization states can be reversibly transformed from sextuple to quadruple to double, which has never been reported in any other ferroelectric systems so far.

Given this improved clarification, we hope the Reviewer would agree that the present work indeed contains substantial new findings to warrant its publication in Nature Communications.

To avoid potential future confusion, we have also added the following sentences on page 8 of the revised manuscript:

“Here we note that the multiple polarization states as established here are fundamentally different from the multiple resistive states reported in previous studies of ferroelectric synapses^{6,8,42-44}, as distinctly contrasted by the observations of mutually convertible six/four/two polarization states in a single hysteresis loop in the former case, and only two polarization states in the latter case”.

[1] Nat. Commun. 13, 699 (2022). (Ref. 6 in the manuscript).

[2] Nat. Commun. 8, 14736 (2017). (Ref. 8 in the manuscript).

[3] Adv. Sci. 9, 2200566 (2022).

[4] Neuromorph. Comput. Eng. 2, 022001(2022).

[5] Phys. Rev. Appl. 19, 014054 (2023).

Comment 1: *The authors mentioned that it is possible to tune polarization orders (Fig.3 and Fig.S9). However, the hysteresis loops appear to distinguish between the original (Fig.3a) and recovered (Fig. 3e) sextuple states. Quadruple cases (Fig. 3b .vs. Fig. 3d; Fig. S9a .vs. Fig. S9c) also exhibit this behavior. I thus doubt whether the polarization states belonging to the initial and subsequent polarization orders are exactly the same ones.*

Response: In principle, ferroelectric polarizations can be switched reversibly to precise states, resulting in perfectly symmetric and reproducible hysteresis loops for ideal materials systems. However, with practical considerations, real samples include various unintended defects “*such as local variations in stoichiometry, vacancies, stacking faults, layer thickness variations, surface defects, and contaminants*”, as itemized in the questions raised by Reviewer 2 (questions 1~3). Because of those defects, the polarization switching details such as the coercive voltage and polarization value can be varied among multiple hysteresis loops, while the same macroscopic state can be formed with different microscopic combinations of electric dipoles. Thus, it is challenging to experimentally observe identically repeatable ferroelectric hysteresis loops as

proposed in theories, even for some traditional ferroelectric materials [6~8]. We also observed hysteresis loops with double polarization states demonstrated similar repeatability to the previously reported double-state loops of CIPS at room temperature [9,10]. Additionally, from the practical application point of view, such as nonvolatile ferroelectric memories, the functionality of a ferroelectric material is maintained as long as the multiple polarization states can be distinguished within an acceptable tolerance.

In response to the Reviewer's comment, we have added the following sentences on page 11 of the revised manuscript:

“We note that the hysteresis loops observed before and after the reversible transformation are not exactly the same, which can be due to various unavoidable defects in the CVT-grown CIPS films during the deposition and subsequent transfer processes, such as local variations in stoichiometry, vacancies, stacking faults, etc. For example, variations in stoichiometry can lead to local phase separation, which induces local strain and helps to stabilize the quadruple polarization states as proposed in Ref 20. Cu vacancies and excess Cu ions have been proposed to have the effect of lowering the energy barrier of Cu to move across the vdW gaps²¹. Stacking faults can lead to local kink, edge-type, or knot-type dislocations, which can cause the formation of nanodomains in CIPS films⁴⁷. Nevertheless, even though the polarization switching details such as the coercive voltage and polarization value can be varied because of the presence of the aforementioned defects, the polarization orders can be distinguished and transformed within an acceptable tolerance”.

[6] Nano Convergence. 1, 24 (2014).

[7] Small Methods 6, 2101289 (2022).

[8] J. Appl. Phys. 108, 042006 (2010).

[9] Appl. Phys. Lett. 109, 172901 (2016).

[10] Adv. Electron. Mater. 6, 2000760 (2020).

Comment 2: *The off-field amplitude hysteresis loops in Fig. 2c around 4-6 V bias window show that the number of minima varies across cycles. The individual stability and ability to retain after withdrawing electric fields of each sextuple polarization state should be carefully discussed including the time scale.*

Response: As elucidated in response to the Reviewer's comment 1, the stability and repeatability of remanent polarization states can be influenced by various types of defects. More importantly, the nonuniform electric field distribution and testing time scale can also affect the resultant formation of the remanent polarization states. It is well known that defects can efficiently pin the domain walls [11~13] and cause variations in the coercive voltage (i.e. the minima in the amplitude loop), compromising the repeatability of the measured hysteresis loops. On top of those structural and chemical defects, the dynamic movement of Cu ions unique to the CIPS system adds additional randomness even under a uniform electric field. That is, when the Cu ions are excited by a tip-induced nonuniform electric field, the repeatability of our CIPS hysteresis loops can be influenced further. We agree with the Reviewer that the number of amplitude minima varies in our samples because of the randomness from defects and dynamic movement of the Cu ions. We have

additional datasets exemplifying a discrete and less varied number of amplitude minima throughout the entire bias window (Fig. R2). Importantly, although certain variability of coercive voltages and polarization values among multiple loops are inevitable in the currently reported CIPS material system, the emergence of sextuple polarization states remains consistent and replicable across various sample batches.

Furthermore, the Reviewer's inquiry regarding the time scale effect is indeed insightful as one of the major causes of variability in the hysteresis loops. To investigate this effect, we conducted additional experiments where we varied the pulse durations for hysteresis loop measurements, as illustrated in Fig. R3. We found that the sextuple remanent polarization states remained consistent across pulse width (both excitation and reading pulses) of 15 ms (used in our manuscript) and 50 ms. This indicates that variations in this time scale range do not alter the major characteristics of the sextuple polarization states.

In response to the Reviewer's comment, Figures R2 and R3 have been added to the Supplementary Information as new Fig. S16 and S17. We have also added the description on pages 8 and 9 as

“This phenomenon has been observed consistently over 23 CIPS samples from various crystal batches (an additional dataset is present in Fig. S16)”.

“Additionally, the stability of the sextuple remanent states has been investigated with varied reading pulse duration (Fig. S17), suggesting that the variations in the time scale of 15~50 ms do not alter the major characteristics of the observed sextuple polarization states”.

Figure R2. Additional dataset showing clear six minima in multiple cycles of amplitude loop.

Figure R3. Ferroelectric hysteresis loops demonstrating sextuple polarization states obtained under pulse widths of 15 and 50 ms. The top row shows the piezoresponse, the middle row shows the phase signal, and the bottom row shows the amplitude signal.

[11] J. Appl. Phys. 129, 014102 (2021).

[12] Adv. Electron. Mater. 3, 1600505 (2017).

[13] Adv. Mater. Interfaces 9, 2101647 (2022).

Comment 3: *Regarding the mechanism of the sextuple-polarization state, the authors attribute it to different polarization configurations depending on the applied electric field. If so, why are there sextuple states rather than other numbers? The authors indicated in Fig. S12 that some configurations have relative energy levels. To effectively demonstrate this point, all polarization configurations should be taken into account and divided into six groups. In this regard, more theoretical calculations are required.*

Response: We apologize for the confusion raised by Fig. S12, caused by insufficient presentation on the related theory part. We would like to clarify that the sextuple states originate from the stacking of three domains but not the polarization states of a multilayer (e.g. 6-ML shown in Fig. S12). According to our theory in a separate and (in fact) preceding study (listed below), stacking three layers into a trilayer with each layer having the intralayer antiferroelectric (AFE) coupling can lead to sextuple states with net polarization (see Fig. R4). The underlying physics is that the AFE ground state of the monolayer harbors a unique half-layer-by-half-layer flipping mechanism. In the present paper, the predicted mechanism is applied to our systems, but the three layers are replaced by three domain blocks, with each domain block consisting of a thin film of CuInP_2S_6 . The primary purpose of Fig. S12 is to illustrate the plausibility of the existence of the AFE coupling within a domain block consisting of six CuInP_2S_6 layers. The AFE ground state is also found for other films, whose thickness is even larger than twenty layers.

More details of the trilayer first-principles study can be found in Ref. 48 of the revised manuscript, which now has been updated as “Yu, G., Pan, A., Zhang, Z. Chen, M. Polarization multistates in antiferroelectric van der Waals materials. arXiv.2312.13856 (2023).”

Figure R4. Multistates in trilayer CuInP_2S_6 . There are six polarization states for a CuInP_2S_6 trilayer, for which a half-layer-by-half-layer flipping mechanism is operative during the transformation between the states.

Comment 4: *The mechanism of the quadruple polarization states has been well explained in ref. 20. Both the HP and LP states share the same polarization arrangements. Their difference is caused by the absence or presence of displacement in Cu sites. To my understanding, the mechanism involving the change in polarization configurations mentioned here is distinct from that. I wonder why did they observe comparable hysteresis loops?*

Response: Indeed, as highlighted by the Reviewer, the mechanism underpinning the change in polarization configurations proposed in our study differs from that reported in Ref. 20. In the context of Ref. 20, despite the observation of four distinct polarization states, each of the hysteresis loops displayed in that earlier study merely illustrates the polarization transition between two of these states (Fig. R5a). The simultaneous manifestation of all four states within a single hysteresis loop was uncharted in Ref 20, but a subsequent study conducted by the same group did show a four-state three-level hysteresis loop in CIPS films via polarization switching among two ferroelectric polarization states and two antiferroelectric states (Fig. R5b) [14]. In contrast, our quadruple-state hysteresis loops observed in the CIPS film with sextuple pristine polarization states (Fig. R5c) demonstrate distinctly different features, including two subloops in a single hysteresis loop that directly correspond to four polarization states, and a polarization direction that aligns antiparallel to the electric field direction when subjected to a sufficiently strong electric field. Such distinctions underscore a clear divergence in mechanisms between the earlier publications [14 and Ref. 20] and our current findings, as also pointed out by the Reviewer. Additionally, it is noteworthy that the team of Ref. 20 also obtained the phenomenon of polarization direction against that of the applied electric field in their later works [15,16]. The intricate mechanism of this novel phenomenon is currently not well understood and warrants further in-depth exploration in the future.

In response to the Reviewer's comment, we added the following sentence on page 14: **“suggesting a distinct mechanism from that proposed in Ref. 20”**.

Figure R5. (a) The hysteresis loop presented in Ref. 20 in the manuscript. (b) Hysteresis loop reported in a later work [14]. (c) Hysteresis loop showing quadruple polarization states in our work. The y-axis piezoelectric constant, d , used in (a) and (b) is proportional to the piezoresponse (c) used in our work [17], thus the trend of the hysteresis loops can be compared among these different works.

[14] ACS Nano 16, 2452 (2022).

[15] Phys. Rev. Appl. 13, 064063 (2020). (Ref 21 in the original manuscript)

[16] Adv. Electron. Mater. 8, 2100810 (2022).

[17] J. Mater. Sci. 41, 107 (2006).

Comment 5: *The reliability of machine-learning potential should be assessed in advance. It is best to provide a force and energy comparison between DFT and DeepMD. Considering that 4-ML CIPS falls inside the DFT range, I suggest combining DFT and machine-learning potential to compare the total energies of all polarization states for 4-ML CIPS.*

Response: We have plotted Fig. R6 for comparison of the energies and forces between DFT and DeepMD calculations. One can see that the results of our DP potential are in good agreement with those of DFT, suggesting the high accuracy of our DP results. Fig. R6 has been added to the Supplementary Information as new Fig. S18. We also added descriptions on page 23 as

“The energies and atomic forces predicted by DP method for all configurations in the test dataset (Fig. S18) agree well with those of DFT, suggesting the high accuracy of the DP results”.

Figure R6. Comparison of energies and atomic forces predicted by DFT and DP model for all configurations in the test dataset.

Detailed Responses to Reviewer 2

Generic Comments: *In this work, the authors experimentally demonstrated that antiferroelectric van der Waals CuInP2S6 films can be stabilized into various polarization states, including double, quadruple, and sextuple polarization states. They also showed that a system with a polarization order of six can be reversibly tuned to an order of four or two, providing a way to control the number of polarization states.*

While the experimental results are interesting and significant, I have several concerns over the explanation and interpretation of the results and the proposed underlying switching mechanisms. Thus, I would like to seek clarifications from the authors.

Response: We are pleased that the Reviewer finds the experimental results interesting and significant. We have addressed the Reviewer's concerns point-by-point in the following responses.

Comment 1: *In CVD-grown CIPS, various defects can occur during the deposition and subsequent transfer processes, for example, local variations in stoichiometry, vacancies, stacking faults, layer thickness variations, surface defects, and contaminants. These defects can significantly impact the ferroelectric domains and their switching behavior. It would be insightful if the authors can assess the impacts of these defects on the domain behavior.*

Response: We must admit that we couldn't agree more with the Reviewer that the CVD-grown CIPS crystal is inevitably prone to various types of defects during the growth and exfoliation processes. Achieving precise control over the stacked domains is challenging due to the presence of these defects, as evidenced by the broad distribution of coercive voltages in Fig. 2e and in many previously reported articles [7, 18~20]. Despite the fluctuations of hysteresis loops caused by those defects, to our pleasant surprise, the AFE-coupled CIPS trilayers proposed in our DFT work (full paper available at arXiv.2312.13856) can be qualitatively extended to our proposed three domain block system that well explained our experimental observations of sextuple polarization states in much thicker CIPS films.

Local defects like variations in stoichiometry, vacancies, and stacking faults in the CIPS system have been reported in the literature. The variations in stoichiometry can lead to local phase separation, which induces local strain and helps to stabilize the quadruple polarization states as proposed in Ref. 20. Cu vacancies and excess Cu ions have been proposed to lower the energy barrier of Cu to move across the vdW gap [15]. The stacking faults can lead to local kink, edge-type, or knot-type dislocations, which can cause the formation of nanodomains in the CIPS films [21]. Beyond those existing studies, the effects of layer thickness variation and surface defects have been assessed directly in our experiments. As an example, Fig. R7 illustrates the correlation between film thickness and imprints based on CIPS films showing double polarization states. We

found that the thinner the film, the smaller the imprint (Fig. R7a), which can be attributed to the reduced AFE domain volume in the subsurface under the excitation of the tip-induced nonuniform electric field. In addition, the imprint value changes from negative to positive when the film thickness is down to 9.1 nm. This observation can be due to the built-in field caused by the charge transfer from the n-doped silicon substrate (surface defects), which becomes prominent in ultrathin CIPS films. The positive imprint and the opposite built-in field have been reported in CIPS using p-doped silicon substrates [22], consistent with our present findings. More interestingly, the trend of imprints with varied thicknesses is found similar to that of the exchange bias of classic ferromagnetic/antiferromagnetic bilayer [23] (Fig. R7b). These detailed findings are to be included in a more systematic publication that is under preparation.

Although our experimental capabilities fall short of achieving exact control over various defects, discrete sextuple polarization states have been unambiguously and repeatably established in our experiments. While recognizing these current limitations, we acknowledge the Reviewer to raise the potential for further optimization in future works. Overall, CIPS is an intriguing material worthy of further exploration. Ultimately, we aim to prepare a CIPS trilayer as proposed by the DFT calculations to validate the sextuple polarization states with controllable defects. Such experiments, once achieved, will provide a much clearer understanding of the effect of defects on domain behaviors.

To discuss the impact of defects, we have added the following sentences on page 11 of the revised manuscript (as in the responses to Reviewer 1):

“We note that the hysteresis loops observed before and after the reversible transformation are not exactly the same, which can be due to various unavoidable defects in the CVT-grown CIPS films during the deposition and subsequent transfer processes, such as local variations in stoichiometry, vacancies, stacking faults, etc. For example, variations in stoichiometry can lead to local phase separation, which induces local strain and helps to stabilize the quadruple polarization states as proposed in Ref 20. Cu vacancies and excess Cu ions have been proposed to have the effect of lowering the energy barrier of Cu to move across the vdW gaps²¹. Stacking faults can lead to local kink, edge-type, or knot-type dislocations, which can cause the formation of nanodomains in CIPS films⁴⁷. Nevertheless, even though the polarization switching details such as the coercive voltage and polarization value can be varied because of the presence of the aforementioned defects, the polarization orders can be distinguished and transformed within an acceptable tolerance”.

Figure R7. Imprint ($V_{\text{ex-FE}}$) variation in CIPS films of different thicknesses. (a) Hysteresis loops demonstrating double polarization states from CIPS films of different thicknesses. The red dotted lines and the extended pink area represent the average imprint (\bar{X}) and the standard deviation σ , respectively. (b) $V_{\text{ex-FE}}$ value versus the CIPS film thickness. The inset shows the trend of $V_{\text{ex-FE}}$ (based on the normalization of 90% saturation value) versus the normalized thickness in IrMn(t)/Co(1 nm)/Pt(2nm) (black curve) and CIPS films (red curve).

[18] Nat. Commun. 2, 591 (2011).

[19] Phys. Rev. Mater. 2(8), 084414 (2018).

[20] Phys. Chem. Chem. Phys. 21(33), 18240-18249 (2019).

[21] ACS Appl. Nano Mater. 3, 8161-8166 (2020).

[22] Adv. Funct. Mater. 32, 2201359 (2022).

[23] Phys. Rev. B 98, 064413 (2018).

Comment 2: *It is known that defects can pin the domain boundaries and control the domain boundary movements. I am wondering how the characteristic times of these pinning and de-pinning processes compare with the DC pulse width and the rise time of each pulse. It would be insightful if the authors can discuss these issues.*

Response: As highlighted by the Reviewer, the phenomena of pinning and depinning of domain walls, primarily induced by defects such as point defects [6, 24] and disorders [25], have been widely discussed in the literature. These defects can pin the domain boundaries and augment the energy barrier associated with polarization switching. For our ferroelectric hysteresis measurements, we set the DC pulse width at 15 ms, incorporating a rise time of 0.5 ms for each pulse. Probed by the Reviewer's comment, we did additional experiments using 50 ms DC pulse width, which demonstrated discrete sextuple polarization states as well (Fig. R3). On the other hand, the direct visualization and quantification of pinning and de-pinning times induced by defects in CIPS are beyond our current experimental capability, and notably, such characteristic times are so far also absent in other published experimental examinations of CIPS. According to the quantum-molecular-dynamics result that suggests Cu movement time of ~ 100 ps range across a vdW gap in CIPS with excess Cu [15], we believe that our pulse time considerably exceeds the characteristic times of the pinning and de-pinning from defects in CIPS. Given the time scale for simulated Cu ion movement [15] and our experimental observation, the influences stemming from the defect-induced pinning and depinning processes are unlikely to skew our measurements.

To further discuss the issue of the characteristic time of defect pinning and applied electric pulses, we have added the following sentences on page 12 of the revised manuscript:

“It is also known that defects can pin the domain boundaries and augment the energy barrier associated with polarization switching. For CIPS systems, the characteristic time associated with the domain wall pinning and depinning from defects have not been experimentally observed yet. According to a quantum-molecular-dynamics result that suggests Cu movement time of ~ 100 ps across a vdW gap in CIPS with excess Cu²¹, we conjecture that our applied electric pulse time (in the millisecond range as described in the Method section)

considerably exceeds the characteristic times of the pinning and de-pinning from defects in CIPS”.

[24] Nat. Commun. 11, 1762 (2020).

[25] Phys. Rev. Mater. 5, 074402 (2021).

Comment 3: *There are large variations in the coercive voltages for different polarization states (Figure 2e). The authors should discuss the physical origins.*

Response: As the Reviewer highlighted in comment 1, different types of defects can exist in the CVT-grown CIPS crystals. These defects can indeed contribute to variations in local coercive voltages measured at different locations and across different samples using Piezoresponse Force Microscopy (PFM) when they interact with domain walls. Many examples of defects-induced variation of coercive voltages have been reported in the literature [26, 27]. In the case of CIPS, in addition to the effect of defects, the variation of coercive voltage can also be attributed to the active Cu movements at room temperature [9].

The data presented in Figure 2e were compiled from our comprehensive dataset, without discriminating between crystal batches, film thicknesses, measurement locations, or ranges of bias windows. It is noteworthy that the standard deviation of coercive voltages, based on hysteresis loops measured on the same sample (std1) (Fig. R8b), exhibits much narrower variations compared to that presented in Figure 2e (std2) (Fig. R8a) as shown in Table R1. We are actively engaged in ongoing efforts to precisely control these defects and plan to comprehensively report on their influence on coercive voltages in our future report.

To discuss the variation of the coercive voltages illustrated in Fig. 2e, we have added the following sentences on page 8 of the revised manuscript:

“We note that there are large variations in the coercive voltages for different polarization states, which can be attributed to the different types of local defects and active Cu movements at room temperature. In addition, the data presented in Fig. 2e were compiled from our comprehensive dataset, without discriminating between crystal batches, film thicknesses, measurement locations, or ranges of bias windows. It is noteworthy that the variation of V_c is much narrower based on the hysteresis loops measured on one sample (Table S3) compared to Fig. 2e”.

Table R1 has also been added to the Supplementary Information as the new Table S3.

Figure R8. Distribution of coercive voltages from hysteresis loops showing sextuple polarization states. (a) the original figure 2e from ensemble-averaged data, (b) distribution from one CIPS film sample.

	-Vc3	-Vc2	-Vc1	+Vc1	+Vc2	+Vc3
Average value from one sample	-5.42	-3.10	-1.76	0.74	3.28	4.90
Standard deviation from one sample (std1)	0.37	0.54	0.32	0.11	0.61	0.60
Average value from multiple samples (Fig. 2e)	-6.13	-3.71	-2.02	1.21	3.88	5.87
Standard deviation from multiple samples (std2)	1.10	0.96	0.59	0.63	1.08	1.29
Ratio of std1/std2	0.34	0.56	0.54	0.17	0.56	0.47

Table R1. Comparison of the average and the standard deviation of coercive voltages measured from one sample and multiple samples showing sextuple polarization states.

[26] Proc. Natl. Acad. Sci. U.S.A.104 (51), 20204 (2007).

[27] ACS Appl. Electron. Mater. 3(2), 619-628 (2021).

Comment 4: *It seems that the authors used ML models to explain quadruple and sextuple polarization states in pure CIPS (the reference is not available for the Reviewer). However, the authors used DBs (~ 10 MLs) to explain these multiple polarization states here. This change has not been properly justified. In my opinion, this is a significant weakness of the present work.*

Response: Again, we apologize for the incompleteness of referring to the relevant theory work in our original submission. In that separate theory work based on first-principles calculations, stacking two MLs of the CIPS family with the intralayer AFE coupling into a bilayer can give rise to quadruple states with net polarization, while stacking three MLs can give rise to sextuple polarization states for a trilayer system. The theory paper now can be accessed via arXiv.2312.13856 (Ref. 48 in the revised manuscript). Encouragingly, and intriguingly, this mechanism can be invoked to explain the sextuple states observed in our experiments for thin films with the assumption of three AFM-coupled ferroelectric domains. The results of a film with 10 MLs are to schematically show the polarization in a domain for different states.

Comment 5: *The discussion on the DBs and their boundary structures and dynamics is absent. It would be insightful if the authors can discuss these issues. Just curious, are the DB boundaries associated with defects?*

Response: We agree with the Reviewer that the domain boundary structures and dynamics are important and warrant attention. We posit that defects (e.g. excess Cu ions, Cu vacancies, random AFE domain spots, etc.) with no net polarization contribution in CIPS can initiate boundaries between domain blocks, which work as “dead layers” between domain blocks. Our DFT work predicted that the quadruple and sextuple polarization states can be formed in pure bilayer and trilayer CIPS films even without these “dead layers” (arXiv.2312.13856). The experimental fabrication and examination of 1~3 ML CIPS with ferroelectricity have not been reported in the field yet, but we expect that the present work will stimulate future studies in more detailed investigation on domain structure modulations in the ultimate few-layer CIPS systems.

To provide more insight into the DB boundary and dynamics, we have added the following sentences on page 15 of the revised manuscript:

“We posit that the defects with no net polarization contribution in CIPS, such as excess Cu ions, Cu vacancies, and random AFE domain spots, can initiate the boundaries between the DBs, which work as “dead interfacial layers”.

Comment 6: *What are the energy barriers between these different states? Can the detection voltages or thermal fluctuations affect the state stability?*

Response: To experimentally observe the energy barriers, constructing an Arrhenius plot that spans a range of temperatures exceeding a decade is typically necessary. Regrettably, such an accepted approach falls beyond our current experimental capabilities. Considering the challenges posed by a large film thickness of CIPS and defect conditions, directly calculating energy barriers of tens of nanometer CIPS films via DFT simulations is hindered by demanding huge computational power. However, our DFT calculations on trilayer CIPS can provide insight into the energy barriers among sextuple polarization states, falling within the range of 283~393 meV (Fig. R9, full paper available at arXiv.2312.13856). Based on the well-known compensation effect, we also speculate that the energy barriers for DB switching are in the range larger than the maximum energy barrier (393 meV), but smaller than the sum of them (676 meV) with lowered attempt frequency.

Figure R9. The kinetic pathways and energy barriers for CIPS trilayers as the systems transform between the six polarization states.

In addition, to address the Reviewer's concerns regarding detection voltages, we have conducted additional experiments of local ferroelectric hysteresis measurements using varied detection voltages at the same location (Fig. R10). When the $V_{ac}=0.2$ V, the small detection voltage leads to a weak response signal and relatively large variability, especially in the range of 2~6 V. $V_{ac}=0.3$ V gives a relatively repeatable signal with reduced variations among multiple loops and improved amplitude response, which was mostly used in our PFM measurements. We found that the sextuple polarization states remain stable when $V_{ac} < 1$ V, but in the range of $0.5 \text{ V} < V_{ac} < 1$ V the repeatability of multiple hysteresis loops gradually deteriorated. The sextuple polarization states are severely distorted when $V_{ac} \geq 1.5$ V. The excessive detection voltage can lead to poor repeatability and disrupt the intermediate states, which may cause the states to collapse into more stable states with deeper potential wells.

Furthermore, we would like to acknowledge the Reviewer for raising the issue of the importance of the thermal effect. We did *in situ* spectroscopic PFM measurements at varied temperatures using the Heater-Cooler attachment of our SPM equipment (Fig. R11). The measurements conducted at room temperature (25 °C) demonstrate the typical hysteresis loops showing sextuple polarization states. When the sample temperature was directly lowered to 5 °C, the classic double-state hysteresis loop was observed with improved repeatability and nearly constant amplitude, and the sextuple polarization states vanished. We expect that at lower temperatures (say 5 °C), the energy barrier heights defining the sextuple polarization states are too high, and the external electric field can only facilitate Cu ions to overcome the two most shallow potential wells, thereby resulting in double polarization states only. When the sample temperature was returned to 25 °C, the sextuple polarization states emerged again and even retained at 40 and 45 °C, until reaching the Curie temperature (~50 °C) where ferroelectricity disappeared. At elevated temperatures, the energy barrier heights defining the sextuple polarization states are lowered, possibly due to more active Cu ions movement. Furthermore, the largely varied amplitude levels at higher temperatures also indicate the complex subsurface domain formation and interaction at the remanent state caused by the rich Cu ion movements, which also result in reduced repeatability of the loops. Importantly, all of the above additional experimental results will not change our main experimental observations described in the manuscript.

In response to the Reviewer's comments, we have added the following sentences on page 9 and page 15 of the revised manuscript, respectively:

“Moreover, appropriate detection voltage is necessary to observe reliable sextuple polarization states. The excessive detection voltage can lead to poor repeatability and disrupt the intermediate states, which may cause the states to collapse into more stable states with deeper potential wells (Fig. S19).”

“Our DFT calculations of the 3-ML CIPS system provide insight into the energy barriers among sextuple polarization states, falling within the range of 283~393 meV⁴⁸. Based on the well-known compensation effect in activation processes, we speculate that the energy barriers for DB switching are in the range larger than the maximum energy barrier (393 meV), but smaller than the sum of them (676 meV) with lowered attempt frequency. The proposed mechanism is further corroborated by the temperature-dependent experiments (Fig. S20). At lower temperatures (5 °C), the energy barrier heights for the sextuple polarization states are too high, and the external electric field can only facilitate Cu ions to overcome the two most shallow wells, thereby resulting in double polarization states only. At elevated temperatures (between room temperature and Curie temperature), the energy barrier heights for sextuple polarization states are readily overcome, which may be due to more active Cu ions movement, resulting in the emergence of sextuple polarization states”.

Figures R10 and R11 have been added to the Supplementary Information as new Fig. S19 and S20.

Figure R10. Effect of varied detection voltage V_{ac} on the hysteresis loop of sextuple polarization states. For each panel, the top, middle, and bottom rows show the piezoresponse, phase, and amplitude signals, respectively.

Figure R11. Temperature-dependent ferroelectric hysteresis loops. For each panel, the top, middle, and bottom rows show the piezoresponse signal, phase signal, and amplitude signal, respectively.

Comment 7: *The authors speculated that the quadruple and sextuple states were possibly due to the nonuniform electric field under the tip. If this is the case, the tip size plays a crucial role in the distribution of the electric field under the tip. But the impact of the tip size was not discussed. If the tip radius decreases from the current 30 nm to, for example, 5 nm, what are the consequences?*

Response: We thank the Reviewer for this insightful proposal. In response, we have first conducted simulations to model the nonuniform electric field distribution under a 5 nm tip using COMSOL Multiphysics (Fig. R12). When compared to the scenario with a 25 nm tip, at the same field strength (e.g. 10^7 V/m, close to the coercive field of CIPS reported in the literature), the field extended distance along both the in-plane (x-axis) and out-of-plane (z-axis) directions is shrunk to half. In other words, the active volume of CIPS under the 5 nm conductive tip is reduced to about 1/6 of that using a 25 nm tip. In addition to the largely shrunk excited volume, the sensible range of the tip is also significantly reduced with the tip radius.

To further investigate the above-mentioned field-gradient effect, we have performed PFM experimental measurements on both a classic ferroelectric ceramic (PMN-PT single crystal) and a CIPS film using a ~ 5 nm radius tip (AD-2.8-SS, Adama, Ireland), which has a conductive diamond coating with similar stiffness and resonant frequency as those of the ~ 25 nm tip (PPP-EFM, Nanosensors, Switzerland) (Fig. R13). The ~ 25 nm tip can detect reliable polarization switching for both materials. Unfortunately, it turns out that the polarization switching was not achieved even in the classic ferroelectric ceramic using a ~ 5 nm tip in our current system. We will continue to investigate the field gradient effect on the intrinsic multiple polarization orders in CIPS systems,

and we speculate that, for a trilayer system, the 5-nm tip should be operative in defining multiple polarization states.

To discuss the impact of tip size, we have added a new section of “Supplementary Note 3” in Supplementary Information. Figures R12 and R13 have been added as Fig. S21 and S22 accordingly.

Figure R12. Electric field distribution in CIPS under a conductive tip of 5 nm radius. (a,c,e) Electric field distribution along the depth direction (Z). (b,d,f) Electric field distribution along the in-plane direction (X).

Figure R13. PFM ferroelectric loops of PMN-PT and CIPS measured using conductive tips with ~ 25 nm and ~ 5 nm radius of curvature, respectively.

Finally, we would like to sincerely thank the two reviewers again for their careful and expertized reviews, as well as for their constructive comments and suggestions. We hope this further revised manuscript is now ready for publication in Nature Communications.

REVIEWER COMMENTS

Reviewer #1 (Remarks to the Author):

The authors have addressed all the concerns I and another reviewer raised in the initial report and have appropriately modified the manuscript. Considering the improvements to the paper after the revision, now I would like to recommend its publication in Nature Communications.

Reviewer #2 (Remarks to the Author):

I would like to thank the authors for taking efforts to address my comments and concerns. However, I am still not totally convinced by their theoretical modelling.

1. The physical properties of the DBs are still not discussed in details. In the model, the theoretical minimum number of atomic layers needed to form a DB is 3. However, in the experiment, the sample with 2 states (hence 1 DB in total) has ~12 atomic layers. This suggests that, in theory, this single DB in the 2-state system should be able to be split into 3 or 4 DBs. The reason that this is not observed experimentally is rather fundamental to the idea of DB and should be further discussed.

2. The model does not describe how the switching between polarization orders is carried out. In the model, the DBs are separated by "dead interfacial layers" and each DB behaves like a unit. However, the model has not explained how the "controllable mutual transformation among polarization orders" can be achieved. For example, when the system switches from 6 states to 4 states by increasing the bias window, how do the original 3 DBs in the 6-state system interact with one another to form the 2 larger DBs in the 4-state system, and how will the "dead layers" be incorporated into the new DB? Furthermore, if the 4-state system is switched back to a 6-state system, are the DBs in the new 6-state system the same as DBs in the original 6-state system? If yes, why; if no, what are the consequences? This is important as it is associated with the robustness of the system to such polarization order switching.

Addressing these concerns would greatly enhance my confidence in recommending the acceptance of the study.

Responses to Reviewers' Second Reports
(MS # NCOMMS-23-52264B by Tao Li et al.)

Again, we thank the two reviewers for their careful and expert review of the above manuscript. The detailed responses are listed below, and the manuscript has been revised accordingly.

Detailed Responses to Reviewer 1

Generic Comments: *The authors have addressed all the concerns I and another reviewer raised in the initial report and have appropriately modified the manuscript. Considering the improvements to the paper after the revision, now I would like to recommend its publication in Nature Communications.*

Response: We thank the Reviewer for assessing that the present manuscript is now acceptable for publication in Nature Communications.

Detailed Responses to Reviewer 2

Generic Comments: *I would like to thank the authors for taking efforts to address my comments and concerns. However, I am still not totally convinced by their theoretical modelling.*

Response: We thank the Reviewer for confirming our efforts to address the comments and concerns. We have addressed the Reviewer's new comments point-by-point in the following responses.

Comment 1: *The physical properties of the DBs are still not discussed in details. In the model, the theoretical minimum number of atomic layers needed to form a DB is 3. However, in the experiment, the sample with 2 states (hence 1 DB in total) has ~12 atomic layers. This suggests that, in theory, this single DB in the 2-state system should be able to be split into 3 or 4 DBs. The reason that this is not observed experimentally is rather fundamental to the idea of DB and should be further discussed.*

Response: We thank the reviewer for finding our experimental results interesting and significant in the first round of review. In terms of the model, we apologize for the unclear writing that raised likely confusion between the theoretical model based on 3-ML CIPS and the definition of DBs. In the experiments, each DB acts in unison like an ML in the modeling paper, irrespective of the specific number of MLs that it contains. According to current experimental observations of the thinnest CIPS films that obtained the sextuple, quadruple, and double polarization states, we speculate that each DB can contain 9~10 MLs as explained in Supplementary Note 1 in

detail, which effectively act as one ML in the theoretical model. Again, we hope a direct confirmation of the predictions using a 3-ML CIPS system can be achieved in the future.

To avoid potential future confusion, we have added the following sentences on page 15 of the revised manuscript:

“The DB is assumed as an energetically metastable elementary unit that effectively acts like an ML in the AFE/FE model and can be collectively modulated by a moderate E-field (as exemplified in Fig. 4c)”.

Comment 2: *The model does not describe how the switching between polarization orders is carried out. In the model, the DBs are separated by "dead interfacial layers" and each DB behaves like a unit. However, the model has not explained how the "controllable mutual transformation among polarization orders" can be achieved. For example, when the system switches from 6 states to 4 states by increasing the bias window, how do the original 3 DBs in the 6-state system interact with one another to form the 2 larger DBs in the 4-state system, and how will the "dead layers" be incorporated into the new DB? Furthermore, if the 4-state system is switched back to a 6-state system, are the DBs in the new 6-state system the same as DBs in the original 6-state system? If yes, why; if no, what are the consequences? This is important as it is associated with the robustness of the system to such polarization order switching.*

Response: The observed mutual transformation among polarization orders is predominantly influenced by the bias window, representing the maximum strength of the electric field serving as an initializing field for the polarization orders. As the maximum field strength increases from that required for the sextuple to the quadruple/double states, we anticipate that a more uniform domain structure can be produced and more defects can be driven to move around. Upon sweeping the electric bias repeatedly, the tip-induced nonuniform electric field generates the corresponding domain structures that form the quadruple/double polarization states. Conversely, electric field sweeping with a reduction in the maximum field strength disturbs the uniform domain structure and the defects, leading to the emergence of sextuple polarization states again. Owing to the defects and/or AFE MLs in the CIPS films, the newly formed sextuple polarization states should not be identical to the original ones, as indicated by variations in coercive biases and polarization values. Nevertheless, the sextuple polarization states remain distinctly discernible.

In response, we have added the following sentences on page 18 of the revised manuscript:

“Furthermore, the mutual transformation among different polarization orders is predominantly influenced by the bias window, representing the maximum strength of the electric field serving as an initializing field for the polarization orders. As the maximum field strength increases from that required for the sextuple to the quadruple/double states, we anticipate that a more uniform domain structure can be produced and more defects can be driven to move around. Upon sweeping the electric bias repeatedly, the tip-induced nonuniform electric field can produce the corresponding domain structures that form the quadruple/double polarization states. Conversely, electric field sweeping with a reduction in

the maximum field strength disturbs the uniform domain structure and the defects, leading to the emergence of sextuple polarization states again”.

Finally, we would like to sincerely thank the two reviewers again for their careful and expertized reviews, as well as for their constructive comments and suggestions. We hope this further revised manuscript is now ready for publication in Nature Communications.

REVIEWERS' COMMENTS

Reviewer #2 (Remarks to the Author):

I believe the authors have done what they do now. I have no other comments.